# LSSInst: Improving Geometric Modeling in Explicit BEV Perception with Instance Representation

## Abstract

With the attention gained by camera-only 3D object detection in autonomous driving, methods based on Bird-Eye-View (BEV) representation especially derived from the explicit lift-splat-shoot (LSS) paradigm, have recently seen significant progress. The BEV representation is ideal for learning the road structure and scene layout. However, to retain computational efficiency, the compressed BEV representation is inevitably weak in retaining the individual geometric details, undermining the methodological generality and applicability. With this in mind, to compensate for the missing details and utilize multi-view geometry constraints, we propose LSSInst, a two-stage object detector incorporating BEV and instance representations in tandem. The proposed detector exploits fine-grained pixel-level features while can be flexibly integrated into existing LSS-based BEV networks. Having said that, due to the inherent gap between two representation spaces, we design the instance adaptor for the BEV-to-instance semantic coherence rather than pass the proposal naively. Extensive experiments demonstrated that our proposed framework is of excellent generalization ability and performance, which boosts the performances of modern LSS-based BEV perception methods without bells and whistles and outperforms current LSS-based state-of-the-art works on the large-scale nuScenes benchmark.

## 1 Introduction

As a crucial component in 3D perception, 3D object detection can be applied in various fields, such as autonomous driving and robotics. Although LiDAR-based 3D detection methods (Vora et al., 2020; Lang et al., 2019; Zhou & Tuzel, 2018; Yan et al., 2018) are verified as having remarkable performance, research in camera-based methods has received increasing attention in recent years. The reasons can be attributed not only to the lower deployment cost but also to the advantages offered by long-range distance and the identification of visual road elements (Li et al., 2022b; Park et al., 2023). However, unlike LiDAR sensors that provide direct and accurate depth information, detecting objects solely based on camera sensor images poses a significant challenge. Thus, how to utilize multi-view images to build up effective representations has become a key issue.

Recently, significant progress has been achieved in methods that utilize the bird's-eye view (BEV) which can be mainly categorized as explicit type (Li et al., 2023b;a; Feng et al., 2023; Park et al., 2023) based on lift-splat-shoot (LSS) (Philion & Fidler, 2020) and implicit type (Li et al., 2022b; Yang et al., 2023) based on the learnable BEV query. Due to its purely implicit aggregation by uninterpretable but forcibly dense queries, the implicit type shows lower performance and expansibility (Han et al., 2023), enabling the explicit LSS-based type to become mainstream in modern BEV paradigms for camera-only 3D detection at present. Based on the LSS hypothesis and the fact that most objects in the scene are close to the ground, LSS-based BEV provides a perspective with minimal parallax ambiguity and information loss in observing the objects as a whole. Illustrated by Fig. 1 (I), these methods look around and gather information from multiple 2D views and create a comprehensive representation of the scene. This representation is in the form of a planar view with compressed height ($z$-axis) and reduced resolution to ensure computational efficiency. The BEV feature benefits from its holistic representation and dense feature space, making it well-suited for capturing the scene's structure and data distribution. However, the geometrically-compressed nature of the BEV

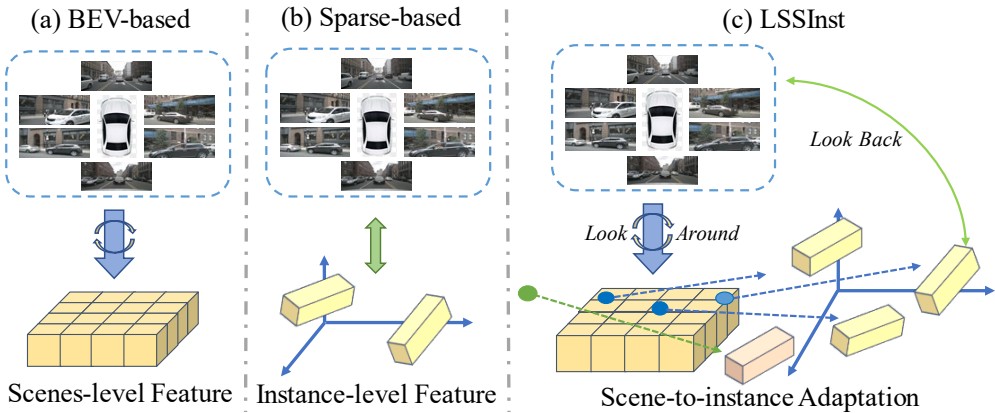

Figure 1: The conceptual comparison of our framework with previous camera-based fashions.

Table 1: The per-category AP comparison between two typical fashions with equivalent detection ability ($\Delta$mAP less than **0.5%**) methods on the nuScenes `test` set.

| | Method | mAP | car | truck | bus | pedestrian | bicycle | traffic_cone | barrier |
|---|---|---|---|---|---|---|---|---|---|
| 1 | BEVDet (Huang et al., 2021) | 42.4 | **0.643** | **0.350** | **0.358** | 0.411 | 0.296 | 0.601 | **0.614** |
| | Spatial-DETR (Doll et al., 2022) | 42.5 | 0.610 | 0.332 | 0.330 | **0.462** | **0.327** | **0.629** | 0.582 |
| 2 | CFT-BEV3D (Jiang et al., 2022) | 41.7 | **0.628** | **0.348** | **0.347** | 0.416 | 0.299 | 0.596 | **0.607** |
| | DETR3D (Wang et al., 2021) | 41.2 | 0.603 | 0.333 | 0.290 | **0.455** | **0.308** | **0.627** | 0.565 |

representation inherently limits its ability to provide precise 3D position descriptions of objects and fully exploit detailed features for object matching particularly in the 3D detection task which requires accurate prediction of 3D object bounding boxes. Meanwhile, as another typical fashion and shown in Fig. 1 (II), sparse-based methods without BEV (Wang et al., 2021; Liu et al., 2022; Chen et al., 2023) leverage instance-level representations and exploit the 3D geometric prior to regress object bounding boxes from the 3D perspective of the objects. However, due to the initialization semantic dispersity, they fail to capture abundant object-aware features from the image at once in comparison with the BEV feature that fits adequate semantic information in the scene, resulting in lower overall performance than the contemporaneous BEV-based methods.

Following this, there are some interesting and corroborative findings in the per-class AP comparison between the two aforementioned fashions as shown in Tab. 1. Notably, considering the practical variety such as data augmentation and training strategies, the difference between the overall mAP values of selected methods in the same group is strictly less than **0.5%** which ensures both detection abilities are equivalent. We can observe that there is the same AP tendency among the classes. Specifically, BEV representation seems more attentive to regular objects (car, bus, truck, barrier) with distinct movements or common positions in the scene, with relative insensitivity to the objects (pedestrian, bicycle, traffic cone) with uncertain trajectories or dispersed locations, which further proves its characteristics of fitting data distribution and leaning to the scene-level focus. Inspired by this, to brighten the complementary synergy of both fashions and make up for the missing details in the representation formulation of current LSS-based BEV perception as well as utilize multi-view geometry constraints, we are motivated to propose **LSSInst**, incorporating the sparse instance-level representations based on the scene-level representations to look back for more detailed feature with geometric matching. As illustrated in Fig. 1 (III), based on the global scene-level pre-feature, instance-level features are pushed to look back at the image locally, focusing on more fine-grained pixel features and allowing for flexible geometric matching, which ultimately generates a final perception result that combines globally-semantic and locally-geometric information.

However, this collaboration also poses challenges, as the most straightforward solution of naively sharing the bounding box proposal is intuitively and experimentally failed [1]. As aforementioned, traditional sparse-based detection methods suffer from initialization semantic dispersity and inadequate

---

[1]See Sec. 4.4 and Tab. 5 for more details

semantic understanding of the scene, the above solution would sever the coherence with the dense representations. With this in mind, we propose the instance adaptor module to establish semantic coherence between the scene and instances and an instance branch for detection. The instance adaptor module generates multiple sparse queries and their corresponding 3D boxes through multi-level adaptive aggregation. The instance branch focuses on fine-grained sparse feature extraction and geometric matching using prepared inputs, such as box embeddings and spatiotemporal sampling and fusion. On the nuScenes dataset, our LSSInst method demonstrates strong generalization ability. Compared to other typical LSS-based methods, LSSInst achieves significant improvements in mAP. Specifically, it outperforms BEVDet by 5.0%, BEVDepth by 2.2%, BEVStereo by 2.6%, and surpasses the state-of-the-art LSS-based method SOLOFusion by 1.6%.

Our main contributions can be concluded as follows: i) We proposed LSSInst, a two-stage framework that improves the geometric details in LSS-based BEV perception with instance representations; ii) we proposed the instance adaptor to maintain the BEV-to-instance semantic coherence and a newly-designed instance branch to look back and aggregate features spatiotemporally for improvement; iii) The proposed framework was verified with great generalization ability and surpassed the state-of-the-art LSS-based methods by extensive experimental results.

## 2    RELATED WORK

### 2.1    LSS-BASED EXPLICIT BEV PERCEPTION

As BEV has proved to be an effective representation for multi-view 3D detection, LSS-based methods that benefit from the explicit formulation process and superior performance become the recent mainstream paradigm. LSS (Philion & Fidler, 2020) is proposed for an end-to-end view transform architecture that lifts images into frustums by predicting depth distribution and splats them into a BEV representation. Then BEVDet (Huang et al., 2021) incorporates exclusive data augmentation techniques for the detection extension. BEVDepth (Li et al., 2023b) and BEVStereo (Li et al., 2023a) improved the depth accuracy by introducing an extra monocular depth network supervised by corresponding LiDAR depth, and multi-view stereo matching between adjacent frames, while BEVDistill (Chen et al., 2022) chose to the model-level distillation from LiDAR. OA-BEV (Chu et al., 2023) and SA-BEV (Zhang et al., 2023a) enhanced the utilization of depth, which integrated the 3D voxel network based on the additional proposal from the 2D detection network and proposed a depth and semantic fusion module respectively for a more enhanced feature. Besides, several works started to perceive the shortage of the current view transformation assumption. AeDet (Feng et al., 2023) introduced the positional compensation for existing coordinate projection while FB-BEV (Li et al., 2023c) integrated a novel forward-backward view transformation module that partially alleviates the projection issues. SOLOFusion (Park et al., 2023) further unified long-term temporal information based on the short-term temporal optimization with Gaussian top-k sampling to boost performance.

Despite these methods making efforts to chase a flawless BEV representation from the LSS process, due to the avoidable depth error and the compressed nature of pooling operations, the yielded BEV representation is weak in retaining the individual geometric details, hence we differently focus on adapting the BEV representation into better geometrical modeling.

### 2.2    INSTANCE-LEVEL REPRESENTATION INTEGRATION IN CAMERA 3D DETECTION

Integrating instance-level representation is ubiquitous in camera 3D detection to enhance perceptual ability. FQNet (Liu et al., 2019) is a three-stage framework for monocular detection that first locally searches for potential boxes and then follows a Fast-RCNN-like way to aggregate the massive object candidate globally for location prediction. Li et al. (2022a) apply a similar way in stereo 3D detection. They first borrow DSGN (Chen et al., 2020) to locally search for possible proposals and then establish the Vernier network to globally form the confidence map based on the stereo pair. For multi-view 3D detection, BEVFormer V2 (Yang et al., 2023) used the perspective network to generate coarse instance features to serve as auxiliary proposals. Unlike they borrow instance-level features in a **bottom-up (i.e., local-to-global)** way, LSSInst uses a totally different **top-down (i.e., global-to-local)** way for improvement.

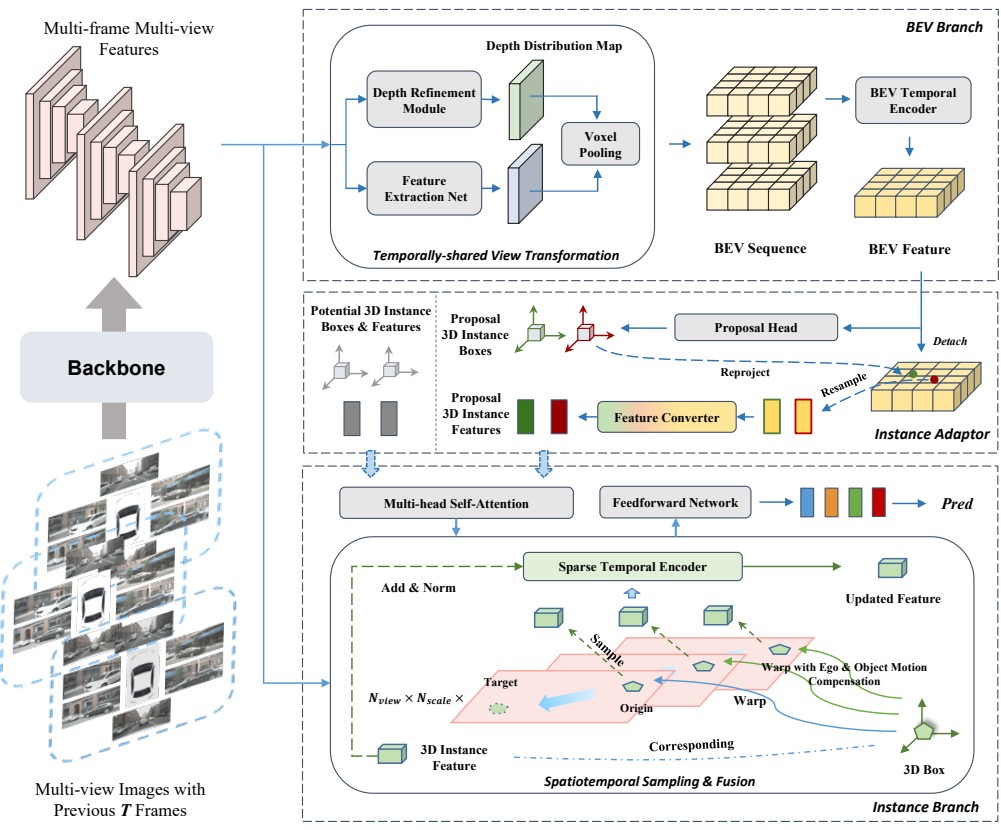

Figure 2: Overview of LSSInst. The multi-view images with previous $T$ frames are fed into the backbone network for the image features. BEV branch looks around the image feature to generate the BEV feature by view transformation and temporal encoding. Instance adaptor aggregates the sparse object-aware feature from the BEV feature and prepares the multiplicate 3D query combination. Instance branch looks back at the image feature and perfects the sparse feature by spatiotemporal sampling and fusion. Lastly, the model makes the final prediction based on the updated output.

## 2.3 TWO-STAGE CAMERA 3D OBJECT DETECTOR

The two-stage design has been widely explored in the 3D detection domain and proved to be effective, whose multi-step workflow is favorable for more accurate prediction. For camera 3D detection, MonoDIS (Simonelli et al., 2022) extracts features from 2D bounding boxes for subsequent 3D bounding box regression. SimMod (Zhang et al., 2023b) utilized a DETR3D head to iteratively refine 2D-level object proposals output from a monocular network. BEVFormer v2 (Yang et al., 2023) extends BEVFormer (Li et al., 2022b) into the second stage by incorporating a first-stage 3D perspective detection network. Among current two-stage methods, they seek more refinement for jointly aggregating coarse samples because their first stage primarily relies on perspective views. Instead, we focus on the subsequent refinement of proposals with holistic semantics derived together from the scene-level layout.

## 3 LSSINST

Utilizing instance-level representations on the basis of the scene-level BEV to excavate more detailed features and geometric information is of practical significance for generalized 3D perception. In this work, we propose LSSInst, which looks back for the more geometry-aware and fine-grained target feature extraction to bridge the adaptation between scene-level and instance-level 3D representations.

The overview of our framework is shown in Fig. 2.3, and we organize the remaining part as follows. Firstly, Sec. 3.1 briefly introduces the BEV branch. Next, Sec. 3.2 introduces the instance adaptor module, and the instance branch is given in Sec. 3.3.

## 3.1 BEV Branch: Looking around for scene-level representation

The multi-view sequential images with the previous $T$ frames are first input into the 2D image backbone network for feature extraction. Then the BEV branch receives the extracted image feature $F_{img} \in \mathbb{R}^{N_v \times N_s \times C \times H \times W}$ and functions as an around looker, translating $F_{img}$ from 2D camera views to BEV for preliminary scene-level representation $F_\beta \in \mathbb{R}^{C \times H_\beta \times W_\beta}$, where $N_v$, $N_s$ denote the camera-view and scale number respectively. This branch can be briefly divided into temporally-shared view transformation for BEV generation and BEV sequence fusion. The 2D-to-BEV view transformation (VT) is naturally based on LSS paradigm, which can be mainly concluded as depth refinement module (DRM), feature extraction net, and voxel pooling. For the best version of the framework, we adopted the Gaussian-spaced top-k stereo (Park et al., 2023) for a better depth distribution map before the voxel pooling. After the shared VT, a sequence of BEV representations will be aligned into current time $t$ and fed to the BEV temporal encoder to form the final current BEV. Here the encoder is designed as a very lightweight residual network for dimension reduction only.

## 3.2 Instance Adaptor: Scene-to-instance adaptation

For the sake of preserving a coherent and solid semantic consistency between BEV and instance representations, we propose the instance adaptor module to eliminate the gaps in the position description and space discrepancy. Since the BEV feature $F_\beta$ is a scene-level representation surrounding the ego car, there is redundancy and inflexibility in modeling instance-level features. To that end, the proposed adaptor module first performs a reprojection of the proposal box coordinates $P_o \in \mathbb{R}^{N_\beta \times 3}$ obtained through the BEV proposal head, returning to the BEV-recognized position $P_\beta \in \mathbb{R}^{N_\beta \times 2}$ to resample the object-related features. Here $N_\beta$ denotes the number of BEV proposals. Given the BEV point-cloud range $R_\beta$ with the corresponding voxel size $S_v$ and up-sampling factor $\sigma$, we can formulate the 2D reprojected coordinate as follow:

$$[P_\beta, z_\beta] = (P_o - R_\beta)/(\sigma S_v) \tag{1}$$

Here, $z_\beta$ denotes the z-axis homogeneous term and is actually a constant of 1. Moreover, due to the overfitting bias in BEV, the focused area may deviate from the actual object position. Inspired by deformable attention (Zhu et al., 2020), the adaptor module incorporates the learnable offsets based on the original focused feature $F_\beta^{'} \in \mathbb{R}^{N_\beta \times C}$ for misalignment compensation by exploring more semantically-aware regions. Suppose $i \in \{1, 2, ..., N_\beta\}$ denotes an arbitrary element index of $F_\beta^{'}$ and its aggregated instance-wise feature $F_\alpha^i$ can be formulated by

$$F_\alpha^i = \mathbf{W}_\alpha \sum_{k=1}^{K} A_k^i \cdot \mathbf{W}_\alpha^{'} \Phi\left(F_\beta, P_\beta^i + \Delta P_{\beta k}^i\right) \tag{2}$$

where $\mathbf{W}_\alpha, \mathbf{W}_\alpha^{'} \in \mathbb{R}^{C \times C}$ are the weight matrix of linear projections and $k$ indexes the resampled keys with the total resampled key number $K$ ($K << H_\beta W_\beta$). $\Delta P_{\beta k}^i \in \mathbb{R}^2$ is the learnable offset and $A_k^i$ is the normalized attention weight of the $k^{th}$ sampling point with $\sum_{k=1}^{K} A_k^i = 1$, which are both linearly projected over $F_\beta^{'}$. $\Phi$ denotes the bilinear interpolation function.

Having said that aforementioned, there still remains the inherent space discrepancy between BEV encoding space and the 3D sparse space suitable for looking back at the image feature. Therefore, we first introduce an extremely-shallow convolutional feature converter $f_{cnv}$ to reparameterize the aggregated features for the inter-space narration. Meanwhile, even with the extensive aggregation and enhancement based on BEV attentive features, a portion of irregular or separated objects cannot be detected due to the BEV overfitting of regular objects and relatively rough perception granularity. Therefore, we introduce extra learnable queries $F_\gamma \in \mathbb{R}^{N_\gamma \times C}$ and reference boxes independent of

BEV proposals, named potential 3D instances and boxes, aimed at capturing the potential BEV-insensitive objects and learning a BEV-agnostic 3D spatial prior. Thus, we can get the multiplicate sparse feature $F_\chi \in \mathbb{R}^{N \times C}$ and here we let $N = N_\beta + N_\gamma$ for simplicity. The whole $F_\chi$ formulation can be derived by

$$F_\chi = \left\{ f_{cnv} \left( \{F_\alpha^i\}_{i=1}^{N_\beta} \right), F_\gamma \right\} \tag{3}$$

### 3.3 INSTANCE BRANCH: LOOKING BACK FOR INSTANCE-LEVEL REPRESENTATION

Given the sequential image features $\{F_{img}^t\}_{t=0}^{T_\chi}$ ($T_\chi \leq T$) from the image backbone network and the sparse instance features $F_\chi$ with corresponding 3D boxes $P_{box} \in \mathbb{R}^{N \times C_{box}}$ from the instance adaptor, the instance branch will spatially and temporally look back the image feature based on the referenced box coordinates and iteratively extract the abundant but more fine-grained representations to update pre-features. This branch can be roughly regarded as a multi-layer Transformer-decoder-like (Vaswani et al., 2017) module for 3D detection, which is briefly divided into two parts: box-level offset and embedding, as well as spatiotemporal sampling and fusion.

**Box-level Offset and Embedding** Different from the previous DETR-like 3D approaches such as DETR3D (Wang et al., 2021), Polorformer (Jiang et al., 2023), VEDet (Chen et al., 2023) that iteratively refine just with 3D coordinate offset regression, the instance branch adopts the box-level offset regression based on $P_{box}$. With this convenience, we can encode all the geometric-aware information of the entire box to substitute the transitional positional encoding, thereby expanding and enriching the space of feature expression rather than the superficial positional level. With it combined with the sparse instance features, there will be more geometric priors and implicit compensation in subsequent attention interactions. Precisely, we first categorize $P_{box}$ based on the element semantic of box dimension into four divisions, which are position $P_{pos} \in \mathbb{R}^{N \times 3}$ (*i.e.*, $x, y, z$), scale $P_{sca} \in \mathbb{R}^{N \times 3}$ (*i.e.*, $w, l, h$), velocity $P_{vel} \in \mathbb{R}^{N \times 2}$ (*i.e.*, $v_x, v_y$), and orientation $P_{ori} \in \mathbb{R}^{N \times 2}$ (*i.e.*, $sin(\theta_{yaw}), cos(\theta_{yaw})$) respectively. Then we introduce five separated linear projections $\{\mathbf{E}_{l3}^j\}_{j=1}^2 \in \mathbb{R}^{3 \times C}$, $\{\mathbf{E}_{l2}^j\}_{j=1}^2 \in \mathbb{R}^{2 \times C}$ and $\mathbf{E}_g \in \mathbb{R}^{C \times C}$ for comprehensive encoding, of which the former four embed every category locally and the last one embeds them globally. The final box embedding $G_\chi \in \mathbb{R}^{N \times C}$ can be formulated by

$$G_\chi = \mathbf{E}_g \left[ \sum_{j=1}^2 \mathbf{E}_{l3}^j \left( P_{d3}^j \right) + \sum_{j=1}^2 \mathbf{E}_{l2}^j \left( P_{d2}^j \right) \right] \tag{4}$$

where $P_{d3}^j, P_{d2}^j$ denote the three and two-dimensional categorized element respectively.

**Spatiotemporal Sampling and Fusion** The sparse feature $F_\chi$ with the box embedding $G_\chi$ will be updated by the spatial and temporal sampling after being fed into the Multi-Head Self-Attention block (Vaswani et al., 2017). Given the corresponding 3D coordinate $P_\chi \in \mathbb{R}^{N \times 3}$ from $P_{box}$, we first warp it from the 3D system to the 2D correspondence $p_\chi \in \mathbb{R}^{N \times 2}$ at the current time by the intrinsic and extrinsic matrix. On the spatial hand, intending to access the target region, we sample the original feature to intermediately regress the existing offset from $p_\chi$ to the target. To expand and voice in the search breadth, we extend the sampling points analogical to Eqn. 2 and enlarge the proportion of residual addition with a weight $\eta$. On the other temporal hand, with time going, there exist ego-car motion and object motion in the autonomous driving scenario, which is required compensation before sampling. In light of the short term in this sparse temporal stereo, *i.e.*, $T_\chi$ is a small positive integer, we approximate the object motion as a uniform rectilinear motion. Thus, we first compensate $p_\chi$ with current velocity $P_{vel}$ and then warp it into every coordinate system as $\{p_\chi^t\}_{t=1}^{T_\chi}$ of per historical time by the medium transition in the global world coordinate system. The per-frame sampled feature $F_{\delta t}, t \in \{0, 1, ..., T_\chi\}$ can be formulated by

$$F_{\delta t} = \mathbf{W}_\chi \sum_{k=1}^K A_{kt} \cdot \mathbf{W}_\chi' \Phi \left[ F_{img}, (\mathbf{M}_t(p_\chi + \tau t \cdot P_{vel}) + \Delta p_{\chi kt}) \right] \tag{5}$$

Table 2: Comparison results of LSS-based and two-stage detectors on 3D detection on the nuScenes `val` set. † denotes the performance without future frames for a fair comparison.

| Method | Backbone | Image Size | mAP↑ | NDS↑ | mATE↓ | mASE↓ | mAOE↓ | mAVE↓ | mAAE↓ |
|---|---|---|---|---|---|---|---|---|---|
| BEVDet (Huang et al., 2021) | ResNet50 | 256 × 704 | 0.283 | 0.350 | 0.773 | 0.288 | 0.698 | 0.864 | 0.291 |
| BEVDet4D (Huang & Huang, 2022) | ResNet50 | 256 × 704 | 0.322 | 0.457 | 0.703 | 0.278 | 0.495 | 0.354 | 0.206 |
| BEVDepth (Li et al., 2023b) | ResNet50 | 256 × 704 | 0.330 | 0.436 | 0.702 | 0.280 | 0.535 | 0.553 | 0.227 |
| BEVStereo (Li et al., 2023a) | ResNet50 | 256 × 704 | 0.346 | 0.452 | 0.659 | 0.277 | 0.550 | 0.498 | 0.228 |
| AeDet (Feng et al., 2023) | ResNet50 | 256 × 704 | 0.358 | 0.473 | 0.655 | 0.273 | 0.493 | 0.427 | 0.216 |
| SA-BEV (Zhang et al., 2023a) | ResNet50 | 256 × 704 | 0.370 | 0.488 | 0.660 | 0.269 | 0.470 | 0.353 | 0.218 |
| FB-BEV (Li et al., 2023c) | ResNet50 | 256 × 704 | 0.378 | 0.498 | 0.620 | 0.273 | 0.444 | 0.374 | 0.200 |
| SOLOFusion (Park et al., 2023) | ResNet50 | 256 × 704 | 0.406 | 0.497 | 0.609 | 0.284 | 0.650 | 0.315 | 0.204 |
| LSSInst | ResNet50 | 256 × 704 | **0.422** | **0.514** | 0.620 | 0.277 | 0.516 | 0.360 | 0.202 |
| SimMOD (Zhang et al., 2023b) | ResNet50 | 800 × 1333 | 0.331 | 0.427 | 0.721 | 0.267 | 0.401 | 0.810 | 0.184 |
| BEVFormer v2 (Yang et al., 2023) † | ResNet50 | 640 × 1600 | 0.388 | 0.498 | 0.679 | 0.276 | 0.417 | 0.403 | 0.189 |
| LSSInst | ResNet50 | 256 × 704 | **0.422** | **0.514** | 0.620 | 0.277 | 0.516 | 0.360 | 0.202 |

where $\tau$ is the time interval between every two adjacent frames, and $\mathbf{M}_t$ is the ego-motion transform matrix from current time to previous $t$ time. Then the multi-frame features are fed into the sparse temporal encoder $f_{enc}$, a naive three-layer MLP, for temporal iterative fusion. Based on our approximation, the projection error will increase with $t$ larger. Thus we let $\lambda$ denote a constant in the range [0, 1], which is introduced for long-term suppression. With the iterative fusion $F_{\delta(t-1)} \leftarrow f_{enc}(\{F_{\delta(t-1)}, \lambda F_{\delta t}\})$, we can get the final sparse sampled feature $f_\delta$ from $\{f_{\delta t}\}_{t=0}^{T_\chi}$. The whole box-level offset $\delta_\chi$ can be calculated as follows:

$$F_\chi \leftarrow F_\chi + \eta F_\delta \qquad \delta_\chi = f_{reg}\{F_\chi + G_\chi\} \tag{6}$$

Here $f_{reg}$ is the box offset regression function of every layer. Notably, we omit the calculation in the scale and view level of $F_{img}$ for simplicity.

## 4 EXPERIMENTS AND ANALYSIS

### 4.1 EXPERIMENTAL SETTINGS

**Dataset** We conducted extensive experiments on the nuScenes 3D detection benchmark (Caesar et al., 2020), a large-scale dataset in the autonomous driving scene. This benchmark consists of 1,000 autonomous driving scenes, with each scene spanning approximately 20 seconds. The dataset is divided into 850 scenes for training (`train`) or validation (`val`) purposes and 150 scenes for testing (`test`). Each frame in the dataset contains six cameras capturing surrounding views, along with a LiDAR-generated point cloud. The dataset provides annotations for up to 1.4 million 3D bounding boxes across 10 different classes.

**Implementation Details** We implemented our network framework utilizing the open-source MMDetection3D (Contributors., 2020) in PyTorch. The learning rate, optimizer, and data augmentation methods used were the same as those in BEVDepth. By default, We set the image size to $256 \times 704$ and utilized ResNet50, pretrained on ImageNet, as the image backbone. The size of the BEV feature in all our experiments was set to $128 \times 128$. Here we set $k = 6$, $T_\chi = 3$, $\eta = 3$. The feature dimension $C$ is 256 and the box dimension $C_{box}$ is 10. The perception ranges for the $X$ and $Y$ axis was [-51.2m, 51.2m], and the resolution of each BEV grid was 0.8m. The time interval $\tau$ is 0.5s, and long-term suppression $\lambda$ is 0.6.

### 4.2 BENCHMARK RESULTS

We compared our approach with LSS-based and two-stage state-of-the-art methods on the nuScenes `val` and `test` sets. The main results are presented in Tab. 2 and Tab. 3 respectively. On the `val` set, we evaluated the performance of LSSInst against other models with the same setting and without the CBGS strategy and future frame usage. The results clearly showcased the superiority of LSSInst, as

Table 3: Comparison results of LSS-based detectors on 3D detection on the nuScenes `test` set. TTA denotes test time augmentation strategy.

| Method | Backbone | Image Size | TTA | mAP↑ | NDS↑ | mATE↓ | mASE↓ | mAOE↓ | mAVE↓ | mAAE↓ |
|--------|----------|-----------|-----|------|------|-------|-------|-------|-------|-------|
| BEVDet (Huang et al., 2021) | V2-99 | 900 × 1600 | ✔ | 0.424 | 0.488 | 0.524 | 0.242 | 0.373 | 0.950 | 0.148 |
| BEVerse (Zhang et al., 2022) | Swin-B | 900 × 1600 | ✔ | 0.393 | 0.531 | 0.541 | 0.247 | 0.394 | 0.345 | 0.129 |
| BEVDet4D (Huang & Huang, 2022) | Swin-B | 900 × 1600 | ✔ | 0.451 | 0.569 | 0.511 | 0.241 | 0.386 | 0.301 | 0.121 |
| OA-BEV (Chu et al., 2023) | V2-99 | 900 × 1600 | ✔ | 0.494 | 0.575 | 0.571 | 0.256 | 0.377 | 0.385 | 0.132 |
| BEVDistill (Chen et al., 2022) | ConvNeXt-B | 640 × 1600 | ✘ | 0.496 | 0.594 | 0.475 | 0.249 | 0.378 | 0.313 | 0.125 |
| BEVDepth (Li et al., 2023b) | ConvNeXt-B | 640 × 1600 | ✘ | 0.520 | 0.609 | 0.445 | 0.243 | 0.352 | 0.347 | 0.127 |
| BEVStereo (Li et al., 2023a) | V2-99 | 640 × 1600 | ✘ | 0.525 | 0.610 | 0.431 | 0.246 | 0.358 | 0.357 | 0.138 |
| AeDet (Feng et al., 2023) | ConvNeXt-B | 640 × 1600 | ✔ | 0.531 | 0.620 | 0.439 | 0.247 | 0.344 | 0.292 | 0.130 |
| TiG-BEV (Huang et al., 2022) | ConvNeXt-B | 640 × 1600 | ✔ | 0.532 | 0.619 | 0.450 | 0.244 | 0.343 | 0.306 | 0.132 |
| SA-BEV (Zhang et al., 2023a) | V2-99 | 640 × 1600 | ✘ | 0.533 | 0.624 | 0.430 | 0.241 | 0.338 | 0.282 | 0.139 |
| FB-BEV (Li et al., 2023c) | V2-99 | 640 × 1600 | ✘ | 0.537 | 0.624 | 0.439 | 0.250 | 0.358 | 0.270 | 0.128 |
| SOLOFusion (Park et al., 2023) | ConvNeXt-B | 640 × 1600 | ✘ | 0.540 | 0.619 | 0.453 | 0.257 | 0.376 | 0.276 | 0.148 |
| LSSInst | ConvNeXt-B | 640 × 1600 | ✘ | **0.546** | **0.629** | 0.464 | 0.251 | 0.341 | 0.265 | 0.120 |

Table 4: Generalization and Geometric-wise Results of LSSInst. (‡ please refer to Footnote 2).

| Method | mAP↑ | NDS↑ | mATE↓ | mASE↓ | mAOE↓ | Param (M) | Training Cost (min/epoch) | Inference Cost (sec/frame) |
|--------|------|------|-------|-------|-------|-----------|--------------------------|---------------------------|
| **BEVDet** (Huang et al., 2021) | 0.260 | 0.319 | 0.830 | 0.292 | 0.758 | 55.7 | 14 | 0.044 |
| **LSSInst** with BEVDet | **0.310** | **0.367** | **0.771** | **0.285** | **0.658** | 64.0 | 18 | 0.051 |
| **BEVDepth4D** (Li et al., 2023b) | 0.343 | 0.458 | 0.691 | 0.281 | 0.610 | 83.5 | 11 | 0.097 |
| **LSSInst** with BEVDepth4D | **0.365** | **0.477** | **0.671** | **0.275** | **0.492** | 91.8 | 13 | 0.109 |
| **BEVStereo** (Li et al., 2023a) | 0.348 | 0.463 | 0.675 | 0.278 | 0.577 | 92.0 | 7 | 0.186 |
| **LSSInst** with BEVStereo | **0.372** | **0.481** | **0.658** | **0.275** | **0.492** | 102.3 | 10 | 0.208 |
| **SOLOFusion** (Park et al., 2023) | 0.406 | 0.497 | 0.609 | 0.284 | 0.650 | 64.4 | 23 | 0.065 |
| **LSSInst** with SOLOFusion | **0.422** | **0.514** | 0.620‡ | **0.277** | **0.516** | 72.8 | 26 | 0.078 |

it outperformed the current LSS-based SOTA, SOLOFusion by a margin of 1.6% in mAP and 1.7% in NDS, and the current two-stage SOTA, BEVFormer v2 by a margin of 3.4% in mAP and 1.6% in NDS. On the `test` set, our LSSInst achieves an mAP of 54.6% and an NDS of 62.9% without any additional augmentation, outperforming all LSS-based methods. Such improvements demonstrate the effectiveness of our LSSInst for improving LSS-based BEV perception with instance representations.

### 4.3 Generalization Ability and Geometric-Wise Boost

To demonstrate the generalization ability of our LSSInst method, we selected prominent LSS-based methods as the BEV branch of LSSInst. The results are presented in Tab. 4. The table reveals that our LSSInst achieves notable improvements in mAP and NDS compared to standalone BEV detectors at a minor cost. In spite of the impressive detection enhancement with 2 5% mAP and NDS, the corresponding costs increase by an acceptable margin. In particular, among all the methods, there is a significant improvement in mATE [2], mASE, and mAOE, indicating that LSSInst can exploit fine-grained pixel-level features and better enhance perceptual capability with aspects of translation, scale, and orientation, which are all relevant in geometric-wise perception.

### 4.4 Ablations and Analyses

This section is the ablation study with analyses. Notably, we select BEVDepth as the BEV branch in the ablation baseline for convenient experimental conduction. For others, please refer to Appendix.

**Multiplicate queries** To further investigate the impact of the multiplicate queries, as shown in Tab. 5, we explored two scenarios: using only proposal queries (referred to as $Q_\beta$) or learnable

---

[2] Actually in the mATE column of Tab. 4, 0.620 mATE in 0.422 mAP also beats 0.609 mATE in 0.402 mAP, please see the experimental verification in Appendix.

Table 5: Query Composition

| Composition of Queries | mAP↑ | NDS↑ |
|---|---|---|
| 450 $Q_\gamma$ | 0.157 | 0.226 |
| 900 $Q_\gamma$ | 0.263 | 0.297 |
| 450 $Q_\beta$ | 0.331 | 0.447 |
| 900 $Q_\beta$ | 0.330 | 0.446 |
| 450 $Q_\beta$ + 450 $Q_\gamma$ | **0.362** | **0.474** |

Table 6: Segmentation mIoU

| Methods | with GT | with baseline |
|---|---|---|
| Baseline | 44.56 | - |
| LSSInst | **46.63** | **66.21** (>50) |

Table 7: Box-level Embedding

| Center | Box | Box w/ BE | mAP↑ | NDS↑ |
|---|---|---|---|---|
| | | | 0.343 | 0.458 |
| ✓ | | | 0.354 | 0.467 |
| | ✓ | | 0.354 | 0.466 |
| | | ✓ | **0.362** | **0.474** |

potential queries (referred to as $Q_\gamma$), and incorporating both queries. We can observe that We can observe that on the one hand, relying solely on the potential queries cannot play a major role, and even utilizing all 900 queries yielded mediocre performance, which shows the slow convergence because of the initialization semantic dispersity without the scene-level information basis from BEV as aforementioned. On the other hand, though the proposal queries from BEV alone can achieve overall good results, adding more queries (from 450 to 900) does not achieve a better improvement, and instead there is even a slight decrease in performance, which proves its overfitting characteristics for the scene and the fact of the neglectful detection of missing objects in the scene. However, When incorporating two kinds of queries, the performance is further improved and reaches a new level, demonstrating the effectiveness of the potential queries in capturing the potential BEV-insensitive objects and learning a BEV-agnostic 3D spatial prior. Through the comparison of results, it can be concluded that these two types of queries play their unique roles, and their inseparable and complementary synergy enables the model to have a comprehensive understanding from the global scene level to the local instances level.

**BEV-to-Instance Semantic Coherence** To confirm the BEV-to-instance semantic coherence, we conduct the relevant experiments in two aspects. Assuming there is only one foreground class, we calculate the mIoU metric of semantic segmentation compared with the ground truth and baseline as shown in Tab. 6. According to the results with the ground truth, LSSInst is observed as having better semantic maintenance than the LSS baseline, which shows the improvement of perceptual capability to the extra BEV-insensitive objects in the scene. When it comes to the mIoU with the LSS baseline, the value 66.21% is over 50% which also indicates the promising BEV-to-instance semantic coherence. More qualitative results can be referred to in the Appendix.

**Box-level embedding** In order to showcase the impact of box-level embedding, we conducted the ablation experiment, and the results are presented in Tab. 7. In this experiment, we compared different approaches: utilizing only the center points (referred to as **Center**) or the bounding boxes (**Box**) predicted in the BEV detection and incorporating the bounding boxes along with their corresponding box embedding (**BE**). We can both find the same increase by a margin, which indicates that there is no difference between the two types of offset regression, excluding the possibility of using Box to bring additional information compared with Center. However, by incorporating the box-level embedding, we observed a further remarkable improvement over center point inheritance alone. This significant improvement clearly demonstrates the encoding of candidate boxes helps enhance the geometric priors of the queries, thereby improving the extraction of detailed object features from the image. This compensates for the limitations of the BEV representation and enables a more comprehensive understanding of instances.

## 5 CONCLUSION

Existing LSS-based methods make efforts to build up a desirable BEV representation, but they ignore its inherent shortage of geometric loss in the formulation, suppressing its generality in 3D perception. In this paper, we propose LSSInst, a two-stage detector that improves the geometric modeling of the BEV perception with instance representation. To address the challenge of the gap between two representation spaces, we propose the instance adaptor to keep the BEV-to-instance semantic coherence. Then a newly-designed instance branch is introduced to look back for fine-grained geometric matching and feature aggregation. Extensive experimental results demonstrated that our framework is of great generalization ability in modern LSS-based BEV perceptions and excellent performance, surpassing the current state-of-the-art works. We hope that our work will inspire further exploration of generalized 3D perception in more complex and fine-grained outdoor-scene tasks.

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
