## A  APPENDIX

The supplementary document is organized as follows:

- Sec. A depicts the 3D geometric projection details of the instance branch.
- Sec. B provides the detailed module network architectures and design rationality.
- Sec. C describes the generation details of BEV proposals.
- Sec. D provides the experimental details and additions of LSSInst.
- Sec. E provide the qualitative results and visualization analysis.

## A  INSTANCE-LEVEL 3D GEOMETRIC PROJECTION

For the 3D position ego coordinates $P_{pos} \in \mathbb{R}^{N \times 3}$ at the current time, below are the detailed multi-view geometric projection for instance-level representations. Firstly, on the **spatial** hand, $P_{pos}$, *i.e.*, $(x, y, z)$ is warped into the camera coordinate system by using the per-view extrinsics $\mathbf{M}_{cam} = [\mathbf{R}|\mathbf{t}] \in \mathrm{SE3}$ and intrinsics as 2D points $p_\chi$, *i.e.*, $(u, v)$ as follows:

$$\begin{bmatrix} x_c \\ y_c \\ z_c \\ 1 \end{bmatrix} = \begin{bmatrix} \mathbf{R} & \mathbf{t} \\ 0^3 & 1 \end{bmatrix} \begin{bmatrix} x \\ y \\ z \\ 1 \end{bmatrix}, \quad z_c \begin{bmatrix} u \\ v \\ 1 \end{bmatrix} = \begin{bmatrix} f_x & 0 & c_x \\ 0 & f_y & c_y \\ 0 & 0 & 1 \end{bmatrix} \begin{bmatrix} x_c \\ y_c \\ z_c \end{bmatrix} \tag{7}$$

where $x_c, y_c, z_c$ are the coordinates in camera system, $f_x, f_y, c_x, c_y$ are camera intrinsic parameters, and $\mathbf{R} \in \mathbb{R}^{3 \times 3}, \mathbf{t} \in \mathbb{R}^{3 \times 1}$ denote the spatial rotation and translation matrices.

Secondly, on the **temporal** hand, we warp $p_\chi$ to the target coordinate system at time $t$, and for the unified expression, the set of target systems includes the current system, *i.e.*, $0 \in \{t\}$. On the basis of Sec. 3.3 and Eqn. 5, below is the detailed formulation of $\mathbf{M}_t$. Given both extrinsic calibration matrices to the world coordinate system $\mathbf{M}_{cur2w}, \mathbf{M}_{tgt2w} \in \mathrm{SE3}$, we can construct the transformation matrix $\mathbf{M}_t$ from the current system to the target one by

$$\mathbf{M}_t = \mathbf{M}_{tgt2w}^{-1} \times \mathbf{M}_{cur2w} = \begin{bmatrix} \mathbf{R}_{tgt} & \mathbf{t}_{tgt} \\ 0^3 & 1 \end{bmatrix} \tag{8}$$

where $\mathbf{R}_{tgt} \in \mathbb{R}^{3 \times 3}, \mathbf{t}_{tgt} \in \mathbb{R}^{3 \times 1}$ denote the overall temporal rotation and translation matrices.

## B  NETWORK ARCHITECTURES

**Feature Converter**   The detailed architecture of the feature converter module is the combination of a $3 \times 3$ kernel-size convolution layer with 1 padding and batch normalization, aiming to learn an inter-space adaptation from resampled BEV feature to sparse instance features. Here we convert the whole BEV feature into the adaptive space at first in practice for implementation convenience, and we give a short proof to show the equivalence as follows:

$$\begin{aligned} \widehat{F}_\alpha &= f_{cnv}\left(\{F_\alpha^i\}_{i=1}^{N_\beta}\right) & \text{(Eqn. 3)} \\ &= f_{cnv}\left(\left\{\Phi\left[F_\beta, P_\beta^i\right]\right\}_{i=1}^{N_\beta}\right) & \text{(Eqn. 2)} \\ &= f_{cnv}\left(\Phi\left[F_\beta, \{P_\beta^i\}_{i=1}^{N_\beta}\right]\right) \\ &= \Phi\left[f_{cnv}\left(F_\beta\right), P_\beta\right] \end{aligned}$$

where $\widehat{F}_\alpha$ denotes the converted feature from the set of $F_\alpha^i$, and we omit the specific resampling multipliers in Eqn. 2 for simplicity.

**Sparse Temporal Encoder**   The specific architecture of the sparse temporal encoder is a naive three-layer MLP with GeLU (Hendrycks & Gimpel, 2016) for sparse temporal fusion from $2C$ to $C$. Below are the detailed procedures in Alg. 1. As shown in the algorithm, when the iterative fusion ends, the accumulated highest order of $\lambda$ will come to $(t-1)$, *i.e.*, the impact equals to a $\lambda^{t-1}$ multiplier for every $F_{\delta t}$, which indeed acts as the desirable long-term suppression.

---

**Algorithm 1** The pseudo-code of sparse temporal fusion

---

**Require:** $T_\chi \in \mathbb{N}^+$, $0 < T_\chi \le T$, $0 < \lambda < 1$
1: $\ t \leftarrow T_\chi$
2: **while** $t \ne 0$ **do**
3: $\quad F_{\delta t} \leftarrow \lambda F_{\delta t}$ $\qquad\qquad\qquad\qquad\qquad\qquad\qquad\quad$ ▷ $F_{\delta t}$ is formulated by Eqn. 5
4: $\quad F_{\delta(t-1)} \leftarrow concat[F_{\delta(t-1)}, F_{\delta t}]$
5: $\quad F_{\delta(t-1)} \leftarrow f_{enc}(F_{\delta(t-1)})$
6: $\quad t \leftarrow t - 1$
7: **end while**

---

## C   PROPOSAL GENERATION

We describe the generation pipeline for BEV proposals from the proposal head in this section. The proposal head can be a very lightweight BEV detection head, like CenterHead (Yin et al., 2021), only for generating the raw BEV proposals with their scores $\{\rho^i, s^i\}_i$. With the non-maximum suppression (NMS) operation with a score threshold, we can obtain the 3D bounding box candidates $C_o$. Here the threshold is set as 0.1. Considering the variable amount of candidates, we re-filter them by top-k as follows, and here k is set as 450, half of the classical total number of 3D queries.

$$C_o := \text{top-}k\left(\text{NMS}\left[\{\rho^i, s^i\}_i\right]\right) \tag{9}$$

Notably, there also exists the possibility that the amount is smaller than $k$. We add the blank padding for the rest, where the position is random with a $\pi/2$ yaw, and both scale and velocity are zero.

## D   EXPERIMENTAL SETTINGS AND EXTENSIONS

**Evaluation Metrics**   For 3D object detection in the nuSense benchmark, our study utilizes a set of official predefined metrics to evaluate the performance of our approach. These metrics include mean Average Precision (mAP), Average Translation Error (ATE), Average Scale Error (ASE), Average Orientation Error (AOE), Average Velocity Error (AVE), Average Attribute Error (AAE), and nuScenes Detection Score (NDS). Different from direct 3D IoU usage, here mAP is based on the BEV center distance and is calculated by averaging over distance thresholds of 0.5m, 1m, 2m, and 4m for ten different classes of objects, including car, truck, bus, trailer, construction vehicle, pedestrian, motorcycle, bicycle, barrier, and traffic cone. In addition to mAP, NDS is a comprehensive metric that takes into account other indicators to assess the overall detection performance. The remaining metrics are designed to measure the precision of positive results in concerned aspects, such as translation, scale, orientation, velocity, and attribute.

### D.1   EXPERIMENTAL SETTINGS

Our implementation is conducted in MMDetection3D (Contributors., 2020) with one NVIDIA A100 40G GPU node. The adoption of data augmentation strategies follows the setting of the BEV branch. Specifically, the augmentation strategies can be random flips along the X and Y axes, random scaling and rotation in a limited range in the image or BEV level. As for the FPN (Lin et al., 2016) before each branch, we follow the settings of BEVDepth (Li et al., 2023b) and DETR3D (Wang et al., 2021), respectively and choose SECONDFPN (Yan et al., 2018) with 128-dimensional output and standard FPN (Lin et al., 2016) with 256-dimensional output. We select AdamW (Loshchilov & Hutter, 2017) as the optimizer and set the learning rate as 2e-4. Notably, in the ablation study, we selected BEVDepth (Li et al., 2023b) as the BEV branch in the ablation baseline for convenient experimental conduction. Here the BEV branch used 1+2 frames and the sparse branch of the ablation baseline didn't use temporal information except for the frame ablation.

Table 8: Comparison results of LSS-based detectors for 3D detection on the nuScenes `val` set. All methods in the table are trained with CBGS.

| Method | Backbone | Image Size | mAP↑ | NDS↑ | mATE↓ | mASE↓ | mAOE↓ | mAVE↓ | mAAE↓ |
|---|---|---|---|---|---|---|---|---|---|
| BEVDet (Huang et al., 2021) | ResNet50 | 256 × 704 | 0.298 | 0.379 | 0.725 | 0.279 | 0.589 | 0.860 | 0.245 |
| BEVDet4D (Huang & Huang, 2022) | ResNet50 | 256 × 704 | 0.322 | 0.457 | 0.703 | 0.278 | 0.495 | 0.354 | 0.206 |
| BEVDepth (Li et al., 2023b) | ResNet50 | 256 × 704 | 0.351 | 0.475 | 0.639 | 0.267 | 0.479 | 0.428 | 0.198 |
| STS (Wang et al., 2023) | ResNet50 | 256 × 704 | 0.377 | 0.489 | 0.601 | 0.275 | 0.450 | 0.446 | 0.212 |
| BEVStereo (Li et al., 2023a) | ResNet50 | 256 × 704 | 0.372 | 0.500 | 0.598 | 0.270 | 0.438 | 0.367 | 0.190 |
| AeDet (Feng et al., 2023) | ResNet50 | 256 × 704 | 0.387 | 0.501 | 0.598 | 0.276 | 0.461 | 0.392 | 0.196 |
| SA-BEV (Zhang et al., 2023) | ResNet50 | 256 × 704 | 0.386 | 0.512 | 0.612 | 0.266 | 0.351 | 0.382 | 0.200 |
| SOLOFusion (Park et al., 2023) | ResNet50 | 256 × 704 | 0.427 | 0.534 | 0.567 | 0.274 | 0.511 | 0.252 | 0.188 |
| LSSInst | ResNet50 | 256 × 704 | **0.429** | **0.537** | 0.595 | 0.281 | 0.423 | 0.273 | 0.202 |

Table 9: The experimental verification results of mATE improvement.

| Method | | mAP↑ | mATE↓ |
|---|---|---|---|
| | Out_Box_Num | | |
| LSSInst | 300 | 42.2 | 0.620 |
| | 250 | 42.2 | 0.619 |
| | 200 | 42.1 | 0.617 |
| | 150 | 42.0 | 0.614 |
| | 100 | **41.4** | **0.608** |
| SOLOFusion (Park et al., 2023) | | 40.6 | 0.609 |

## D.2 EXPERIMENTAL EXTENSIONS

In this section, we conducted the experimental extensions to show more persuasive performance results and ablation. To be specific, these results are involved in CBGS (Zhu et al., 2019) and the ablations of whistles and bells.

### D.2.1 PERFORMANCE EXTENSION

CBGS strategy as an incremental trick is popular in several works to further increase model performance. In order to further compare with the LSS-based state-of-the-art methods trained with CBGS, we conducted a performance evaluation in the nuScenes val set. As shown in Tab. 8, our LSSInst achieves an mAP of 42.9% and an NDS score of 53.8%, outperforming all existing methods. These results further demonstrate the missing details improvement and inherent effectiveness of our method despite the class imbalance compensation using CBGS.

### D.2.2 VERIFICATION FOR TRANSLATION IMPROVEMENT

The mA*E is designed to measure a property (here we use * to denote this) by the mean statistical error, and it's actually based on the predicted instances, which does not consider the confidence threshold. Considering that we introduce more queries to capture the missing objects, it also means we are more likely to yield lower mATE for those low-score predictions. In practice, we enhance the confidence level and decrease the output box number to show the mATE variation as shown in the table below. When we change to the 100 output number, we can easily observe the better mATE as well as higher mAP.

### D.2.3 RESULTS OF CATEGORY-LEVEL IMPROVEMENT

This section shows the per-class comparison results between SOLOFusion and LSSInst on the nuScene `val` and `test` set. As illustrated in Fig. 3, we can observe the BEV-insensitive categories such as the traffic cone and bicycle, especially pedestrian have been detected with a remarkable

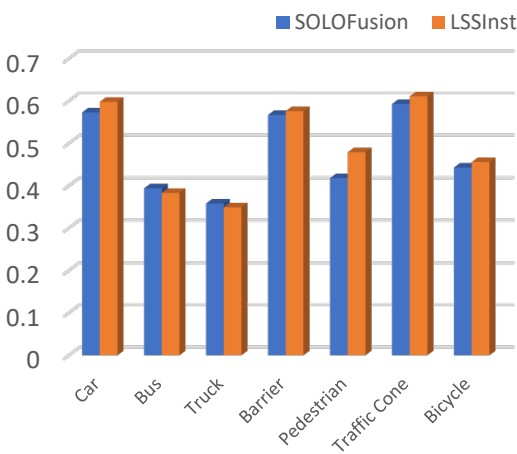

Figure 3: Comparison results of per-classes mAP on nuScenes `val` set.

Table 10: The ablation results of the frame-level extension.

| Frame | mAP↑ | NDS↑ | mATE↓ | mASE↓ | mAOE↓ | mAVE↓ | mAAE↓ |
|---|---|---|---|---|---|---|---|
| BEV Only | 0.366 | 0.477 | 0.661 | 0.278 | 0.625 | 0.327 | 0.205 |
| 1 | 0.381 | 0.494 | 0.662 | 0.271 | 0.473 | 0.358 | 0.207 |
| 2 | 0.382 | 0.495 | 0.654 | 0.271 | 0.470 | 0.360 | 0.207 |
| 3 | **0.389** | **0.497** | 0.652 | 0.270 | 0.454 | 0.378 | 0.223 |
| 4 | 0.383 | 0.496 | 0.659 | 0.273 | 0.462 | 0.366 | 0.198 |

margin. It's favorable for the improvement of the classes with variable movements or dispersed locations since there is a large proportion of human beings (pedestrians) in the auto-driving scenario.

### D.2.4 ABLATION EXTENSION OF WHISTLES AND BELLS

The ablation study below reveals the role and function of each component in our framework. Notably, we select BEVDepth as the BEV branch in the ablation baseline for convenient experimental conduction. The sparse branch of the ablation baseline does not use temporal information except for the frame ablation.

**Offline Temporal Sampling**  Here we particularly change the frame from 3 to 4 for a more comprehensive observing range. As shown in Tab. 10, the results reveal a fluctuating trend. The performance improves gradually as the number of frames increases up to 3, but when the number of frames reaches 4, the performance starts to decline, reflecting a bottleneck. This observation not only indicates that our geometric-guided temporal fusion is helpful for short-term matching and alignment but also shows the theoretical long-term error and verifies the limited approximation mentioned in Sec. 3.3 even though adding the suppression. It can be inferred that as the look-back window increases longer, the objects have moved a larger distance within the interval of much more than $3{\sim}4\tau = 1.5{\sim}2$ seconds, and the variable movement makes it challenging to align the features under short-term geometric constraints, leading to a continuous decrease in performance. In the future, we will adopt online temporal sampling to acquire a wider temporal range to improve the problem.

**Spatial Sampling and Fusion**  As for spatial sampling, we utilize deformable attention to aggregate features from multiple sampling points. As shown in Tab. 11, when we increase the number of sampling points to 2, there is a 0.4% improvement in mAP, indicating that richer spatial sampling helps enrich features and optimize intermediate representations. However, further increasing the number of sampling points results in a performance decline, which may be owing to the smaller resolution of the feature map. As shown in Tab. 12, We explore the performance of different weights

| Table 11: Point Ablation | | |
|:---:|:---:|:---:|
| Points | mAP↑ | NDS↑ |
| 1 | 0.365 | 0.477 |
| 2 | **0.369** | 0.478 |
| 4 | 0.361 | 0.472 |
| 6 | 0.364 | **0.479** |

| Table 12: Weight Ablation | | |
|:---:|:---:|:---:|
| Weight | mAP↑ | NDS↑ |
| 1 | 0.365 | 0.477 |
| 2 | **0.370** | 0.478 |
| 3 | 0.366 | **0.480** |
| 4 | 0.362 | 0.474 |

| Table 13: Adaptor Ablation | | | |
|:---:|:---:|:---:|:---:|
| BDR | FC | mAP↑ | NDS↑ |
| | | 0.3623 | 0.4741 |
| ✓ | | 0.3647 | 0.4769 |
| | ✓ | 0.3651 | 0.4753 |
| ✓ | ✓ | **0.3661** | **0.4779** |

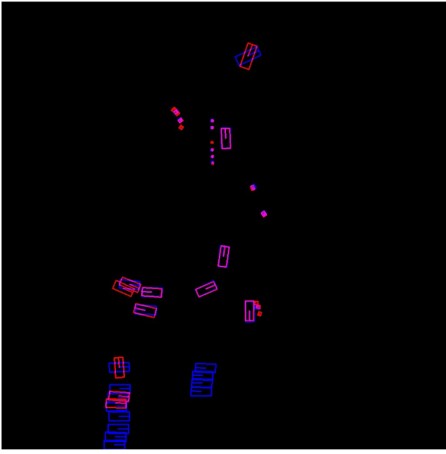 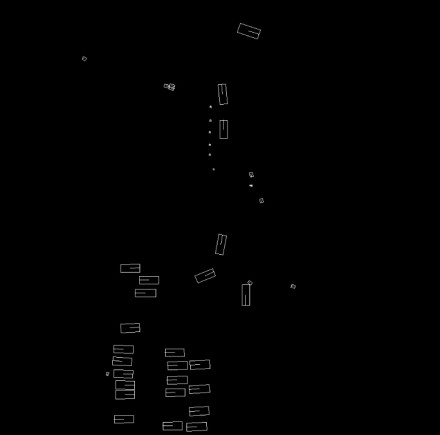

Figure 4: Qualitative comparison between baseline proposals (red), predictions (blue), their superposition (purple), and GT (white).

assigned to image features. The results reveal that increasing the weight of image features to 2 leads to a 0.5% improvement in mAP. This indicates that increasing the weight of image features during spatiotemporal sampling helps enhance the representation ability of queries. The network tends to utilize a larger weight of image features, which further verifies the effectiveness of our designed instance branch for improving intermediate representations.

**Instance Adaptor**    To showcase the effectiveness of the instance adaptor module in LSSInst, we conducted a series of ablation experiments, as depicted in Tab. 13. In this table, **BDR** denotes the BEV feature deformable resampling, and **FC** represents the feature converter. The results indicate that both sub-modules achieved a 0.2% improvement in both mAP and NDS compared to the baseline. When combined, they contributed to a total improvement of 0.4%. This indicates that our instance adaptor module effectively preserves the semantic coherence between BEV and instance representations, enabling effective improvement of BEV features using instance-level information.

## E    QUALITATIVE RESULTS

### E.1    QUALITATIVE COMPARISON OF BEV-TO-INSTANCE COHERENCE

Despite the semantic segmentation mIoU result between LSSInst and the baseline is 66.21% which indicates that our method possesses a desirable semantic scene-layout basis and keeps better semantic coherence, to illustrate this point apparently, here visualize the comparison results between proposals and predictions. It can be more clearly observed not only the coherence but also extra improvement on the basis. As shown in Fig. 4, where blue is yielded by BEVInst, red denotes the proposals, purple means their superposition, and white means GT. We can first conclude that purple boxes occupy the majority. Then there are many red boxes for false or missed detection and some blue boxes for orientation correction or additional detection which match the white boxes much more, which directly proves the improvement.

In this section, we show the visualization comparison results for 3D object detection among LSSInst, ground truth, and current SOTA method SOLOFusion. As shown in Fig. 5, LSSInst has a higher recall and detects more inapparent and occluded objects. For example, our model successfully detects distant cars and trucks in the CAM_FRONT_LEFT and CAM_FRONT_RIGHT views, especially the vehicle occluded by trees and the inapparent car with dark color which is highly similar with the background. Significantly, as the yellow arrow shown in the CAM_FRONT_RIGHT view, we surprisingly find the pedestrian, who is so tiny and indistinct that he/she is even ignored by the ground truth, is captured by LSSInst. Besides, our methods yield a more consistent orientation and box scale with the ground truth in every view. In contrast, for example, there is a severe rotation shift (the red curved arrow) of the bus both in the CAM_FRONT and CAM_FRONT_LEFT views as well as the box misalignment among those cars that are turning past the left traffic lights in the CAM_FRONT_RIGHT view. These observations above fully demonstrate the improvement of missing details, no matter the wider-range perception breadth or own more refined property.

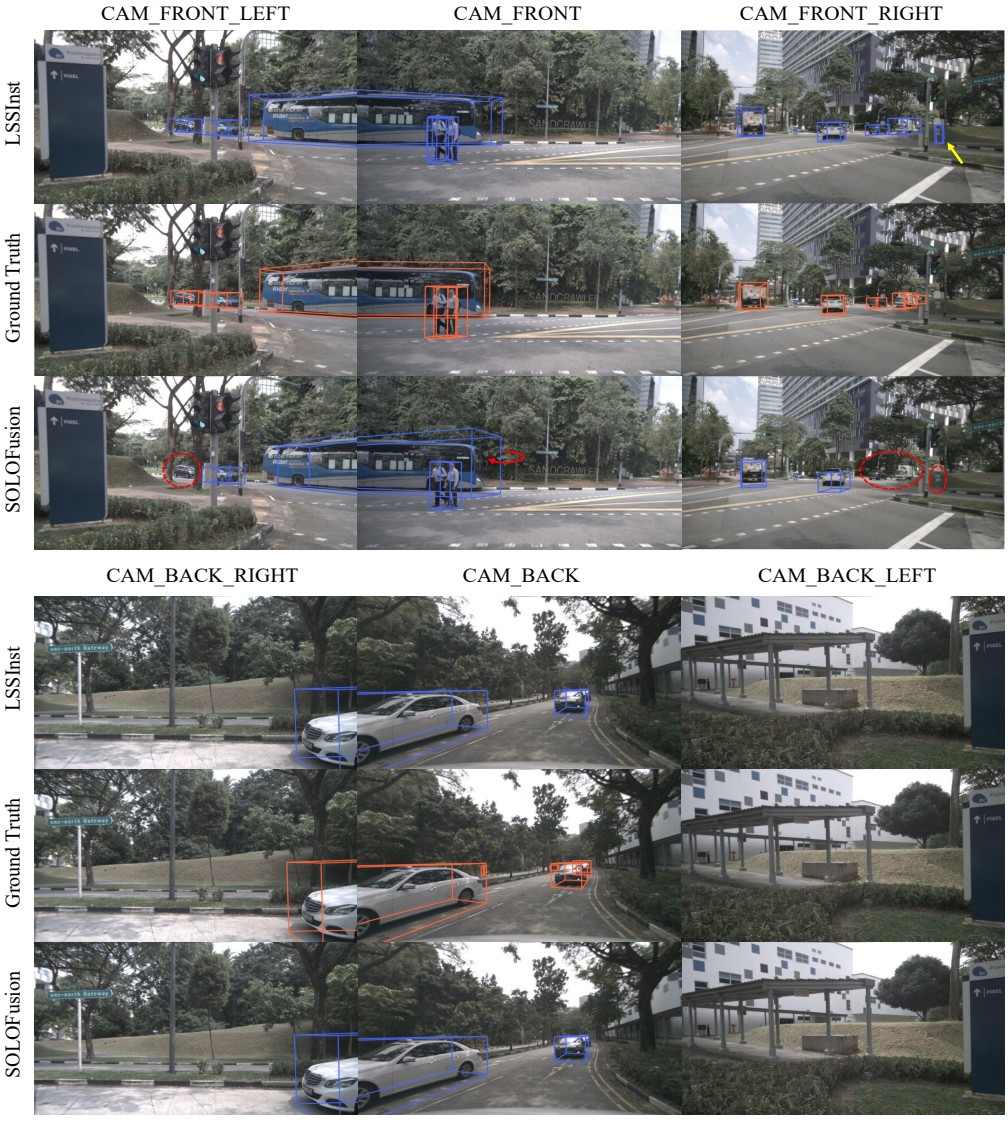

Figure 5: Comparision of LSSInst, the ground truth, and SOLOFusion on nuScenes val set.