# OpenReview forum: "LSSInst: Improving Geometric Modeling in LSS-Based BEV Perception with Instance Representation"
_ICLR.cc/2024/Conference — ICLR 2024 Conference Withdrawn Submission_

### Official Review · Reviewer_TXwr · 2023-10-24

**Soundness:** 2 fair
**Presentation:** 3 good
**Contribution:** 2 fair
**Rating:** 5
**Confidence:** 5

**Summary:**

This work proposes a two-stage BEV detection method, which first generates proposals on BEV space, and then refines the results by projecting the candidate boxes to image planes to resample visual features. The idea sounds reasonable, and the results have demonstrated the effectiveness of the proposed framework. The writing is pretty easy to follow.

**Strengths:**

The proposed method sounds technically reasonable.

The writing is good.

The results show the effectiveness of the introduced method.

**Weaknesses:**

Some concerns are expected to be addressed by the authors,
1) The training cost increases for 30%-50% and the inference speed decreases by more than 10% (see Tab.4 of the manuscript), which is an undeniable burden for realizing such a complex method. Could the authors provide more details about the bottleneck? And is there any plan to speed the framework up?
2) The idea is reasonable but somewhat incremental. Two-stage detectors have been thoroughly proved to be superior to the one-stage counterparts in many fields. It is suggested to provide more explanation about the novelty of this paper.
3) It is confused why encoding box-level information to the network can bring such marginal improvements (see Tab.7 of the paper). In my view, the sampling process in the second stage is based on the predicted box of the first stage, the box-encoding should not show such big influences. Any discussion?

**Questions:**

It seems that the proposed framework can also be applied to implicit-type methods like BEVFormer. It is suggested to provide the results on these methods.

---

### Official Review · Reviewer_ocap · 2023-10-29

**Soundness:** 3 good
**Presentation:** 3 good
**Contribution:** 3 good
**Rating:** 3
**Confidence:** 4

**Summary:**

The authors propose a two-stage detector LSSInst to learn better 3D space representations.
The paper aims to leverage both the advantages of dense forward projection (LSS)  and sparse backward projection (DETR3D). The experimental results also show that the proposed method can improve the performance of existing methods.

**Strengths:**

1. The proposed two-staged design can bring gain to many existing methods.
2. The ablation study is well-designed.

**Weaknesses:**

1. Based on Table 4, the cost of better performance is the lower efficiency.  The reviewer is not excited by this trade-off.
2. Recent sparse-query methods like Streampetr, SparseBEV, and Spase4D methods show that sparse queries alone can achieve higher results with faster inference speed. This has to make the reviewer think about the practical significance of the method proposed in this paper. Since the one-stage detector is better in both performance and efficiency, why do we need a more complex two-stage method?

**Questions:**

See weakness 2

**Details Of Ethics Concerns:**

/

---

### Official Review · Reviewer_hgcA · 2023-11-01

**Soundness:** 2 fair
**Presentation:** 2 fair
**Contribution:** 2 fair
**Rating:** 5
**Confidence:** 5

**Summary:**

This paper proposes a two-stage object detector which simultaenously considers both BEV and instance representation to capture instance-level and scene-level information. They design a BEV-to-instance adator to effective transfer information from BEV to instance representation space. Experiments on the nuScene dataset verify the effectiveness of the proposed method.

**Strengths:**

1. The motivation of this paper makes sense. Table 1 well supports the benefits of fusing BEV and instance features.
2. The performance is strong. It can improve exising methods and achieves good results in various settings.

**Weaknesses:**

1. Even though the motivation can support the method, I think this paper lacks deeper analysis about why BEV-level and instance-level features lead to the performance difference in Table 1. Is this due to the size of the object? I do not quite agree with the regular/irregular objects. More analysis should be conducted, as this is the key insight of this paper.
2. The proposed method is seems a simple combination of BEV-based and instance-based methods. I do not see any non-trivial design that essentially makes the combiniation work. It seems directly combining the two types of methods can lead to improvement. Ablation study about this is missing.
3. It is not clear the necessity of the proposed BEV-to-instance adaptor. No relevant experiments are presented.

**Questions:**

See weakness.

---

### Official Review · Reviewer_6PoT · 2023-11-03

**Soundness:** 3 good
**Presentation:** 3 good
**Contribution:** 2 fair
**Rating:** 5
**Confidence:** 4

**Summary:**

The author of this paper presents a novel approach that combines the strengths of explicitly modeling dense BEV features and sparse query-based methods without modeling BEV. This method results in improvements across various LSS-based methods，and surpasses the state-of-the-art LSS-based method SOLOFusion by 1.6% in mAP. In this paper, extensive comparisons are made between the proposed method and existing LSS-based methods. Additionally, numerous ablation experiments are conducted to demonstrate the effectiveness of the proposed approach.

**Strengths:**

This paper introduces a novel method, refining sparse queries (similar to DETR3D) on dense BEV features. The authors claim that this combination improves the geometric details in LSS-based BEV perception with instance representations. This approach has achieved promising results on the NuScenes dataset, compared to other typical LSS-based methods, LSSInst achieves improvements in mAP. Specifically, it outperforms BEVDet by 5.0%, BEVDepth by 2.2%, BEVStereo by 2.6%, and surpasses the state-of-the-art LSS-based method SOLOFusion by 1.6%

**Weaknesses:**

The title of this paper is "LSSInst." And the paper emphasizes its improvement upon the LSS-based method. In the experimental section, the paper compares its results only with the sota LSS-based methods. However, upon closer examination of the methodology, it can be observed that the core contribution of this paper does not lie in the BEV Branch. In other words, the method proposed in this paper is not necessarily linked to the LSS-based methods, and replacing the BEV Branch with a query-based method would not affect the core aspects of this paper. To demonstrate the generalizability of the proposed method in this paper, it would be valuable for the authors to include experiments where the BEV Branch is replaced with a query-based method. After all, the LSS-based sota SoloFusion has been ranked beyond the top 20 in the NuScenes leaderboard.

**Questions:**

1.	As I mentioned above, to demonstrate the generalizability of the method in this paper, please show results that replace the BEV Branch with query-based methods, like bevformerv2
2.	In the motivation of this paper, it is mentioned that "BEV representation seems more attentive to regular objects (car, bus, truck, barrier) with distinct movements or common positions in the scene, with relative insensitivity to the objects (pedestrian, bicycle, traffic cone) with uncertain trajectories or dispersed locations", and this point is substantiated by the results presented in Table 1. However, this viewpoint is incorrect. Taking BEVFormer-pure and PETR-e as examples, their results on the nuscenes test dataset are as follows, contrary to what is stated in the paper.


	                    map 	     car	truck	bus	    pedestrian	bicycle	traffic_cone	barrier
BEVFormer-pure   0.445	   0.633	0.346	0.335      0.492	 0.36	0.664	         0.587
PETR-e	             0.441	  0.631	0.364	0.344	0.475	0.306	0.66	                 0.608


3.	The result of bevformerv2 in Table 2 is lower than its github repo. The paper mentions the utilization of three temporal frames, while BEVFormerV2 reports the results for two temporal frames for 51.8 NDS and 42.0 mAP, BEVFormerV2 achieves better results than the approach proposed in this paper by using only two temporal frames. Although the proposed method in this paper uses a lower resolution, when both methods are employed at the maximum resolution, the results indicate that the approach in this paper still falls short compared to BEVFormerV2 on the nuscenes test dataset, which achieves 0.58 mAP and 0.648 NDS